# Latent Support Measure Machines
# for Bag-of-Words Data Classification

**Yuya Yoshikawa**
Nara Institute of Science and Technology
Nara, 630-0192, Japan
yoshikawa.yuya.yl9@is.naist.jp

**Tomoharu Iwata**
NTT Communication Science Laboratories
Kyoto, 619-0237, Japan
iwata.tomoharu@lab.ntt.co.jp

**Hiroshi Sawada**
NTT Service Evolution Laboratories
Kanagawa, 239-0847, Japan
sawada.hiroshi@lab.ntt.co.jp

## Abstract

In many classification problems, the input is represented as a set of features, e.g., the bag-of-words (BoW) representation of documents. Support vector machines (SVMs) are widely used tools for such classification problems. The performance of the SVMs is generally determined by whether kernel values between data points can be defined properly. However, SVMs for BoW representations have a major weakness in that the co-occurrence of different but semantically similar words cannot be reflected in the kernel calculation. To overcome the weakness, we propose a kernel-based discriminative classifier for BoW data, which we call the latent support measure machine (latent SMM). With the latent SMM, a latent vector is associated with each vocabulary term, and each document is represented as a distribution of the latent vectors for words appearing in the document. To represent the distributions efficiently, we use the kernel embeddings of distributions that hold high order moment information about distributions. Then the latent SMM finds a separating hyperplane that maximizes the margins between distributions of different classes while estimating latent vectors for words to improve the classification performance. In the experiments, we show that the latent SMM achieves state-of-the-art accuracy for BoW text classification, is robust with respect to its own hyper-parameters, and is useful to visualize words.

## 1  Introduction

In many classification problems, the input is represented as a set of features. A typical example of such features is the bag-of-words (BoW) representation, which is used for representing a document (or sentence) as a multiset of words appearing in the document while ignoring the order of the words. Support vector machines (SVMs) [1], which are kernel-based discriminative learning methods, are widely used tools for such classification problems in various domains, e.g., natural language processing [2], information retrieval [3, 4] and data mining [5]. The performance of SVMs generally depends on whether the kernel values between documents (data points) can be defined properly. The SVMs for BoW representation have a major weakness in that the co-occurrence of different but semantically similar words cannot be reflected in the kernel calculation. For example, when dealing with news classification, 'football' and 'soccer' are semantically similar and characteristic words for football news. Nevertheless, in the BoW representation, the two words might not affect the computation of the kernel value between documents, because many kernels, e.g., linear, polynomial and

Gaussian RBF kernels, evaluate kernel values based on word co-occurrences in each document, and 'football' and 'soccer' might not co-occur.

To overcome this weakness, we can consider the use of the low rank representation of each document, which is learnt by unsupervised topic models or matrix factorization. By using the low rank representation, the kernel value can be evaluated properly between documents without shared vocabulary terms. Blei et al. showed that an SVM using the topic proportions of each document extracted by latent Dirichlet allocation (LDA) outperforms an SVM using BoW features in terms of text classification accuracy [6]. Another naive approach is to use vector representation of words learnt by matrix factorization or neural networks such as word2vec [7]. In this approach, each document is represented as a set of vectors corresponding to words appearing in the document. To classify documents represented as a set of vectors, we can use support measure machines (SMMs), which are a kernel-based discriminative learning method on distributions [8]. However, these low dimensional representations of documents or words might not be helpful for improving classification performance because the learning criteria for obtaining the representation and the classifiers are different.

In this paper, we propose a kernel-based discriminative learning method for BoW representation data, which we call the *latent support measure machine* (latent SMM). The latent SMMs assume that a latent vector is associated with each vocabulary term, and each document is represented as a distribution of the latent vectors for words appearing in the document. By using the kernel embeddings of distributions [9], we can effectively represent the distributions without density estimation while preserving necessary distribution information. In particular, the latent SMMs map each distribution into a reproducing kernel Hilbert space (RKHS), and find a separating hyperplane that maximizes the margins between distributions from different classes on the RKHS. The learning procedure of the latent SMMs is performed by alternately maximizing the margin and estimating the latent vectors for words. The learnt latent vectors of semantically similar words are located close to each other in the latent space, and we can obtain kernel values that reflect the semantics. As a result, the latent SMMs can classify unseen data using a richer and more useful representation than the BoW representation. The latent SMMs find the latent vector representation of words useful for classification. By obtaining two- or three-dimensional latent vectors, we can visualize relationships between classes and between words for a given classification task.

In our experiments, we demonstrate the quantitative and qualitative effectiveness of the latent SMM on standard BoW text datasets. The experimental results first indicate that the latent SMM can achieve state-of-the-art classification accuracy. Therefore, we show that the performance of the latent SMM is robust with respect to its own hyper-parameters, and the latent vectors for words in the latent SMM can be represented in a two dimensional space while achieving high classification performance. Finally, we show that the characteristic words of each class are concentrated in a single region by visualizing the latent vectors.

The latent SMMs are a general framework of discriminative learning for BoW data. Thus, the idea of the latent SMMs can be applied to various machine learning problems for BoW data, which have been solved by using SVMs: for example, novelty detection [10], structure prediction [11], and learning to rank [12].

## 2   Related Work

The proposed method is based on a framework of support measure machines (SMMs), which are kernel-based discriminative learning on distributions [8]. Muandet et al. showed that SMMs are more effective than SVMs when the observed feature vectors are numerical and dense in their experiments on handwriting digit recognition and natural scene categorization. On the other hand, when observations are BoW features, the SMMs coincide with the SVMs as described in Section 3.2. To receive the benefits of SMMs for BoW data, the proposed method represents each word as a numerical and dense vector, which is estimated from the given data.

The proposed method aims to achieve a higher classification performance by learning a classifier and feature representation simultaneously. Supervised topic models [13] and maximum margin topic models (MedLDA) [14] have been proposed based on a similar motivation but using different approaches. They outperform classifiers using features extracted by unsupervised LDA. There

are two main differences between these methods and the proposed method. First, the proposed method plugs the latent word vectors into a discriminant function, while the existing methods plug the document-specific vectors into their discriminant functions. Second, the proposed method can naturally develop non-linear classifiers based on the kernel embeddings of distributions. We demonstrate the effectiveness of the proposed model by comparing the topic model based classifiers in our text classification experiments.

## 3 Preliminaries

In this section, we introduce the kernel embeddings of distributions and support measure machines. Our method in Section 4 will build upon these techniques.

### 3.1 Representations of Distributions via Kernel Embeddings

Suppose that we are given a set of $n$ distributions $\{\mathbb{P}_i\}_{i=1}^n$, where $\mathbb{P}_i$ is the $i$th distribution on space $\mathcal{X} \subset \mathbb{R}^q$. The kernel embeddings of distributions are to embed any distribution $\mathbb{P}_i$ into a reproducing kernel Hilbert space (RKHS) $\mathcal{H}_k$ specified by kernel $k$ [15], and the distribution is represented as element $\mu_{\mathbb{P}_i}$ in the RKHS. More precisely, the element of the $i$th distribution $\mu_{\mathbb{P}_i}$ is defined as follows:

$$\mu_{\mathbb{P}_i} := \mathbb{E}_{\mathbf{x} \sim \mathbb{P}_i}[k(\cdot, \mathbf{x})] = \int_{\mathcal{X}} k(\cdot, \mathbf{x}) d\mathbb{P}_i \in \mathcal{H}_k, \tag{1}$$

where kernel $k$ is referred to as an *embedding kernel*. It is known that element $\mu_{\mathbb{P}_i}$ preserves the properties of probability distribution $\mathbb{P}_i$ such as mean, covariance and higher-order moments by using *characteristic kernels* (e.g., Gaussian RBF kernel) [15]. In practice, although distribution $\mathbb{P}_i$ is unknown, we are given a set of samples $\mathbf{X}_i = \{\mathbf{x}_{im}\}_{m=1}^{M_i}$ drawn from the distribution. In this case, by interpreting sample set $\mathbf{X}_i$ as empirical distribution $\hat{\mathbb{P}}_i = \frac{1}{M_i} \sum_{m=1}^{M_i} \delta_{\mathbf{x}_{im}}(\cdot)$, where $\delta_{\mathbf{x}}(\cdot)$ is the Dirac delta function at point $\mathbf{x} \in \mathcal{X}$, empirical kernel embedding $\hat{\mu}_{\mathbb{P}_i}$ is given by

$$\hat{\mu}_{\mathbb{P}_i} = \frac{1}{M_i} \sum_{m=1}^{M_i} k(\cdot, \mathbf{x}_{im}) \in \mathcal{H}_k, \tag{2}$$

which can be approximated with an error rate of $||\hat{\mu}_{\mathbb{P}_i} - \mu_{\mathbb{P}_i}||_{\mathcal{H}_k} = O_p(M_i^{-\frac{1}{2}})$ [9].

### 3.2 Support Measure Machines

Now we consider learning a separating hyper-plane on distributions by employing support measure machines (SMMs). An SMM amounts to solving an SVM problem with a kernel between empirical embedded distributions $\{\hat{\mu}_{\mathbb{P}_i}\}_{i=1}^n$, called *level-2 kernel*. A level-2 kernel between the $i$th and $j$th distributions is given by

$$K(\hat{\mathbb{P}}_i, \hat{\mathbb{P}}_j) = \langle \hat{\mu}_{\mathbb{P}_i}, \hat{\mu}_{\mathbb{P}_j} \rangle_{\mathcal{H}_k} = \frac{1}{M_i M_j} \sum_{g=1}^{M_i} \sum_{h=1}^{M_j} k(\mathbf{x}_{ig}, \mathbf{x}_{jh}), \tag{3}$$

where kernel $k$ indicates the embedding kernel used in Eq. (2). Although the level-2 kernel Eq.(3) is linear on the embedded distributions, we can also consider non-linear level-2 kernels. For example, a Gaussian RBF level-2 kernel with bandwidth parameter $\lambda > 0$ is given by

$$K_{\mathrm{rbf}}(\hat{\mathbb{P}}_i, \hat{\mathbb{P}}_j) = \exp\left(-\frac{\lambda}{2}||\hat{\mu}_{\mathbb{P}_i} - \hat{\mu}_{\mathbb{P}_j}||_{\mathcal{H}_k}^2\right) = \exp\left(-\frac{\lambda}{2}(\langle \hat{\mu}_{\mathbb{P}_i}, \hat{\mu}_{\mathbb{P}_i} \rangle_{\mathcal{H}_k} - 2\langle \hat{\mu}_{\mathbb{P}_i}, \hat{\mu}_{\mathbb{P}_j} \rangle_{\mathcal{H}_k} + \langle \hat{\mu}_{\mathbb{P}_j}, \hat{\mu}_{\mathbb{P}_j} \rangle_{\mathcal{H}_k})\right).$$
$$\tag{4}$$

Note that the inner-product $\langle \cdot, \cdot \rangle_{\mathcal{H}_k}$ in Eq. (4) can be calculated by Eq. (3). By using these kernels, we can measure similarities between distributions based on their own moment information.

The SMMs are a generalization of the standard SVMs. For example, suppose that a word is represented as a one-hot representation vector with vocabulary length, where all the elements are zero except for the entry corresponding to the vocabulary term. Then, a document is represented by adding the one-hot vectors of words appearing in the document. This operation is equivalent to using a linear kernel as its embedding kernel in the SMMs. Then, by using a non-linear kernel as a level-2 kernel like Eq. (4), the SMM for the BoW documents is the same as an SVM with a non-linear kernel.

## 4 Latent Support Measure Machines

In this section, we propose latent support measure machines (latent SMMs) that are effective for BoW data classification by learning latent word representation to improve classification performance.

The SMM assumes that a set of samples from distribution $\mathbb{P}_i$, $\mathbf{X}_i$, is observed. On the other hand, as described later, the latent SMM assumes that $\mathbf{X}_i$ is unobserved. Instead, we consider a case where BoW features are given for each document. More formally, we are given a training set of $n$ pairs of documents and class labels $\{(d_i, y_i)\}_{i=1}^n$, where $d_i$ is the $i$th document that is represented by a multiset of words appearing in the document and $y_i \in \mathcal{Y}$ is a class variable. Each word is included in vocabulary set $\mathcal{V}$. For simplicity, we consider binary class variable $y_i \in \{+1, -1\}$. The proposed method is also applicable to multi-class classification problems by adopting one-versus-one or one-versus-rest strategies as with the standard SVMs [16].

With the latent SMM, each word $t \in \mathcal{V}$ is represented by a $q$-dimensional latent vector $\mathbf{x}_t \in \mathbb{R}^q$, and the $i$th document is represented as a set of latent vectors for words appearing in the document $\mathbf{X}_i = \{\mathbf{x}_t\}_{t \in d_i}$. Then, using the kernel embeddings of distributions described in Section 3.1, we can obtain a representation of the $i$th document from $\mathbf{X}_i$ as follows: $\hat{\mu}_{\mathbb{P}_i} = \frac{1}{|d_i|} \sum_{t \in d_i} k(\cdot, \mathbf{x}_t)$.

Using latent word vectors $\mathbf{X} = \{\mathbf{x}_t\}_{t \in \mathcal{V}}$ and document representation $\{\hat{\mu}_{\mathbb{P}_i}\}_{i=1}^n$, the primal optimization problem for the latent SMM can be formulated in an analogous but different way from the original SMMs as follows:

$$\min_{\mathbf{w}, b, \xi, \mathbf{X}, \theta} \frac{1}{2} ||\mathbf{w}||^2 + C \sum_{i=1}^n \xi_i + \frac{\rho}{2} \sum_{t \in \mathcal{V}} ||\mathbf{x}_t||_2^2 \quad \text{subject to} \quad y_i \left( \langle \mathbf{w}, \mu_{\mathbb{P}_i} \rangle_{\mathcal{H}} - b \right) \geq 1 - \xi_i, \ \xi_i \geq 0, \ (5)$$

where $\{\xi_i\}_{i=1}^n$ denotes slack variables for handling soft margins. Unlike the primal form of the SMMs, that of the latent SMMs includes a $\ell_2$ regularization term with parameter $\rho > 0$ with respect to latent word vectors $\mathbf{X}$. The latent SMM minimizes Eq. (5) with respect to the latent word vectors $\mathbf{X}$ and kernel parameters $\theta$, along with weight parameters $\mathbf{w}$, bias parameter $b$ and $\{\xi_i\}_{i=1}^n$.

It is extremely difficult to solve the primal problem Eq. (5) directly because the inner term $\langle \mathbf{w}, \mu_{\mathbb{P}_i} \rangle_{\mathcal{H}}$ in the constrained conditions is in fact calculated in an infinite dimensional space. Thus, we solve this problem by converting it into an another optimization problem in which the inner term does not appear explicitly. Unfortunately, due to its non-convex nature, we cannot derive the dual form for Eq. (5) as with the standard SVMs. Thus we consider a min-max optimization problem, which is derived by first introducing Lagrange multipliers $\mathbf{A} = \{a_1, a_2, \cdots, a_n\}$ and then plugging $\mathbf{w} = \sum_{i=1}^n a_i \hat{\mu}_{\mathbb{P}_i}$ into Eq (5), as follows:

$$\min_{\mathbf{X}, \theta} \max_{\mathbf{A}} L(\mathbf{A}, \mathbf{X}, \theta) \quad \text{subject to} \quad 0 \leq a_i \leq C, \ \sum_{i=1}^n a_i y_i = 0, \tag{6a}$$

$$\text{where} \ L(\mathbf{A}, \mathbf{X}, \theta) = \sum_{i=1}^n a_i - \frac{1}{2} \sum_{i=1}^n \sum_{j=1}^n a_i a_j y_i y_j K(\hat{\mathbb{P}}_i, \hat{\mathbb{P}}_j; \mathbf{X}, \theta) + \frac{\rho}{2} \sum_{t \in \mathcal{V}} ||\mathbf{x}_t||_2^2, \tag{6b}$$

where $K(\hat{\mathbb{P}}_i, \hat{\mathbb{P}}_j; \mathbf{X}, \theta)$ is a kernel value between empirical distributions $\hat{\mathbb{P}}_i$ and $\hat{\mathbb{P}}_j$ specified by parameters $\mathbf{X}$ and $\theta$ as is shown in Eq. (3).

We solve this min-max problem by separating it into two partial optimization problems: 1) maximization over $\mathbf{A}$ given current estimates $\bar{\mathbf{X}}$ and $\bar{\theta}$, and 2) minimization over $\mathbf{X}$ and $\theta$ given current estimates $\bar{\mathbf{A}}$. This approach is analogous to *wrapper methods* in multiple kernel learning [17].

**Maximization over A.** When we fix $\mathbf{X}$ and $\theta$ in Eq. (6) with current estimate $\bar{\mathbf{X}}$ and $\bar{\theta}$, the maximization over $\mathbf{A}$ becomes a quadratic programming problem as follows:

$$\max_{\mathbf{A}} \sum_{i=1}^n a_i - \frac{1}{2} \sum_{i=1}^n \sum_{j=1}^n a_i a_j y_i y_j K(\hat{\mathbb{P}}_i, \hat{\mathbb{P}}_j; \bar{\mathbf{X}}, \bar{\theta}) \quad \text{subject to} \quad 0 \leq a_i \leq C, \ \sum_{i=1}^n a_i y_i = 0, \tag{7}$$

which is identical to solving the dual problem of the standard SVMs. Thus, we can obtain optimal $\mathbf{A}$ by employing an existing SVM package.

Table 1: Dataset specifications.

| | # samples | # features | # classes |
|---|---|---|---|
| WebKB | 4,199 | 7,770 | 4 |
| Reuters-21578 | 7,674 | 17,387 | 8 |
| 20 Newsgroups | 18,821 | 70,216 | 20 |

**Minimization over X and $\theta$.** When we fix $\mathbf{A}$ in Eq. (6) with current estimate $\bar{\mathbf{A}}$, the min-max problem can be replaced with a simpler minimization problem as follows:

$$\min_{\mathbf{X},\theta} l(\mathbf{X},\theta), \text{ where } l(\mathbf{X},\theta) = -\frac{1}{2}\sum_{i=1}^{n}\sum_{j=1}^{n}\bar{a}_i\bar{a}_jy_iy_jK(\hat{\mathbb{P}}_i,\hat{\mathbb{P}}_j;\mathbf{X},\theta) + \frac{\rho}{2}\sum_{t\in\mathcal{V}}||\mathbf{x}_t||_2^2. \quad (8)$$

To solve this problem, we use a quasi-Newton method [18]. The quasi-Newton method needs the gradient of parameters. For each word $m \in \mathcal{V}$, the gradient of latent word vector $\mathbf{x}_m$ is given by

$$\frac{\partial l(\mathbf{X},\theta)}{\partial \mathbf{x}_m} = -\frac{1}{2}\sum_{i=1}^{n}\sum_{j=1}^{n}\bar{a}_i\bar{a}_jy_iy_j\frac{\partial K(\hat{\mathbb{P}}_i,\hat{\mathbb{P}}_j;\mathbf{X},\theta)}{\partial \mathbf{x}_m} + \rho\mathbf{x}_m, \quad (9)$$

where the gradient of the kernel with respect to $\mathbf{x}_m$ depends on the choice of kernels. For example, when choosing a embedding kernel as a Gaussian RBF kernel with bandwidth parameter $\gamma > 0$: $k_\gamma(\mathbf{x}_s,\mathbf{x}_t) = \exp(-\frac{\gamma}{2}||\mathbf{x}_s - \mathbf{x}_t||_{\mathcal{H}_k}^2)$, and a level-2 kernel as a linear kernel, the gradient is given by

$$\frac{\partial K(\hat{\mathbb{P}}_i,\hat{\mathbb{P}}_j;\mathbf{X},\theta)}{\partial \mathbf{x}_m} = \frac{1}{|d_i||d_j|}\sum_{s\in d_i}\sum_{t\in d_j}k_\gamma(\mathbf{x}_s,\mathbf{x}_t) \times \begin{cases} \gamma(\mathbf{x}_t - \mathbf{x}_s) & (m = s \wedge m \neq t) \\ \gamma(\mathbf{x}_s - \mathbf{x}_t) & (m = t \wedge m \neq s) \\ \mathbf{0} & (m = t \wedge m = s). \end{cases}$$

As with the estimation of $\mathbf{X}$, kernel parameters $\theta$ can be obtained by calculating gradient $\frac{\partial l(\mathbf{X},\theta)}{\partial \theta}$. By alternately repeating these computations until dual function Eq. (6) converges, we can find a local optimal solution to the min-max problem.

The parameters that need to be stored after learning are latent word vectors $\mathbf{X}$, kernel parameters $\theta$ and Lagrange multipliers $\mathbf{A}$. Classification for new document $d^*$ is performed by computing $y(d^*) = \sum_{i=1}^{n} a_iy_iK(\hat{\mathbb{P}}_i,\hat{\mathbb{P}}_*;\mathbf{X},\theta)$, where $\hat{\mathbb{P}}_*$ is the distribution of latent vectors for words included in $d^*$.

## 5 Experiments with Bag-of-Words Text Classification

**Data description.** For the evaluation, we used the following three standard multi-class text classification datasets: WebKB, Reuters-21578 and 20 Newsgroups. These datasets, which have already been preprocessed by removing short and stop words, are found in [19] and can be downloaded from the author's website[1]. The specifications of these datasets are shown in Table 1. For our experimental setting, we ignored the original training/test data separations.

**Setting.** In our experiments, the proposed method, latent SMM, uses a Gaussian RBF embedding kernel and a linear level-2 kernel. To demonstrate the effectiveness of the latent SMM, we compare it with several methods: MedLDA, SVD+SMM, word2vec+SMM and SVMs. MedLDA is a method that jointly learns LDA and a maximum margin classifier, which is a state-of-the-art discriminative learning method for BoW data [14]. We use the author's implementation of MedLDA[2]. SVD+SMM is a two-step procedure: 1) extracting low-dimensional representations of words by using a singular value decomposition (SVD), and 2) learning a support measure machine using the distribution of extracted representations of words appearing in each document with the same kernels as the latent SMM. word2vec+SMM employs the representations of words learnt by word2vec [7] and uses them for the SMM as in SVD+SMM. Here we use pre-trained 300 dimensional word representation vectors from the Google News corpus, which can be downloaded from the author's website[3]. Note that word2vec+SMM utilizes an additional resource to represent the latent vectors for words unlike the

[1] http://web.ist.utl.pt/acardoso/datasets/
[2] http://www.ml-thu.net/~jun/medlda.shtml
[3] https://code.google.com/p/word2vec/

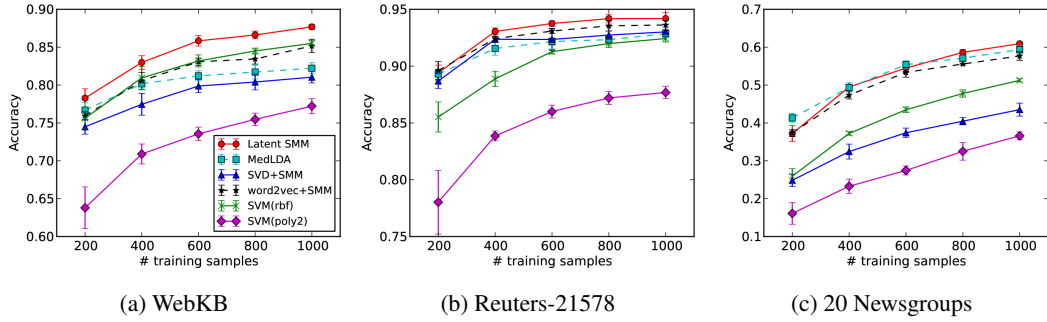

| (a) WebKB | (b) Reuters-21578 | (c) 20 Newsgroups |

Figure 1: Classification accuracy over number of training samples.

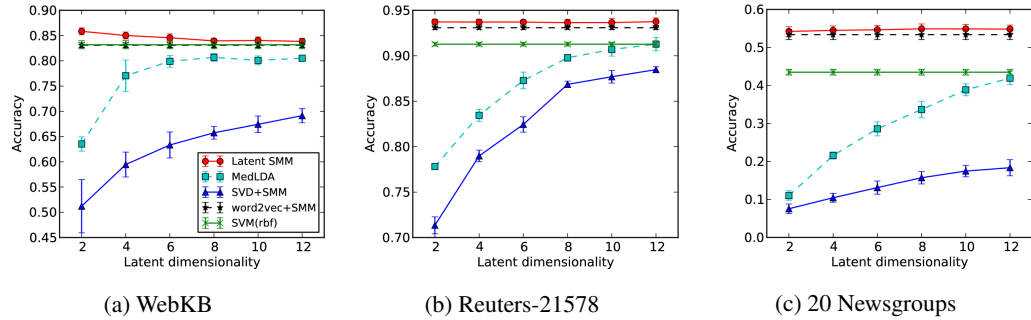

| (a) WebKB | (b) Reuters-21578 | (c) 20 Newsgroups |

Figure 2: Classification accuracy over the latent dimensionality.

latent SMM, and the learning of word2vec requires n-gram information about documents, which is lost in the BoW representation. With SVMs, we use a Gaussian RBF kernel with parameter $\gamma$ and a quadratic polynomial kernel, and the features are represented as BoW. We use LIBSVM[4] to estimate Lagrange multipliers $\mathbf{A}$ in the latent SMM and to build SVMs and SMMs. To deal with multi-class classification, we adopt a one-versus-one strategy [16] in the latent SMM, SVMs and SMMs. In our experiments, we choose the optimal parameters for these methods from the following variations: $\gamma \in \{10^{-3}, 10^{-2}, \cdots, 10^3\}$ in the latent SMM, SVD+SMM, word2vec+SMM and SVM with a Gaussian RBF kernel, $C \in \{2^{-3}, 2^{-1}, \cdots, 2^5, 2^7\}$ in all the methods, regularizer parameter $\rho \in \{10^{-2}, 10^{-1}, 10^0\}$, latent dimensionality $q \in \{2, 3, 4\}$ in the latent SMM, and the latent dimensionality of MedLDA and SVD+SMM ranges $\{10, 20, \cdots, 50\}$.

**Accuracy over number of training samples.** We first show the classification accuracy when varying the number of training samples. Here we randomly chose five sets of training samples, and used the remaining samples for each of the training sets as the test set. We removed words that occurred in less than 1% of the training documents. Below, we refer to the percentage as a word occurrence threshold. As shown in Figure 1, the latent SMM outperformed the other methods for each of the numbers of training samples in the WebKB and Reuters-21578 datasets. For the 20 Newsgroups dataset, the accuracies of the latent SMM, MedLDA and word2vec+SMM were proximate and better than those of SVD+SMM and SVMs.

The performance of SVD+SMM changed depending on the datasets: while SVD+SMM was the second best method with the Reuters-21578, it placed fourth with the other datasets. This result indicates that the usefulness of the low rank representations by SVD for classification depends on the properties of the dataset. The high classification performance of the latent SMM for all of the datasets demonstrates the effectiveness of learning the latent word representations.

**Robustness over latent dimensionality.** Next we confirm the robustness of the latent SMM over the latent dimensionality. For this experiment, we changed the latent dimensionality of the latent SMM, MedLDA and SVD+SMM within $\{2, 4, \cdots, 12\}$. Figure 2 shows the accuracy when varying the latent dimensionality. Here the number of training samples in each dataset was 600, and the word occurrence threshold was 1%. For all the latent dimensionality, the accuracy of the latent SMM was consistently better than the other methods. Moreover, even with two-dimensional latent

[4]http://www.csie.ntu.edu.tw/~cjlin/libsvm/

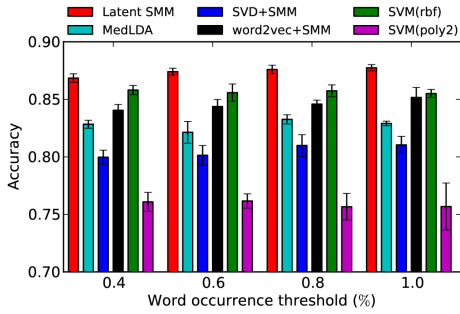
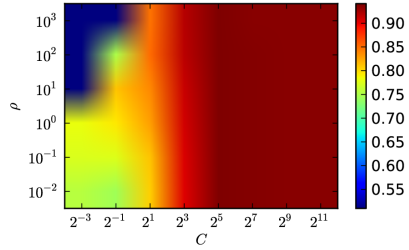

Figure 3: Classification accuracy on WebKB when varying word occurrence threshold.

Figure 4: Parameter sensitivity on Reuters-21578.

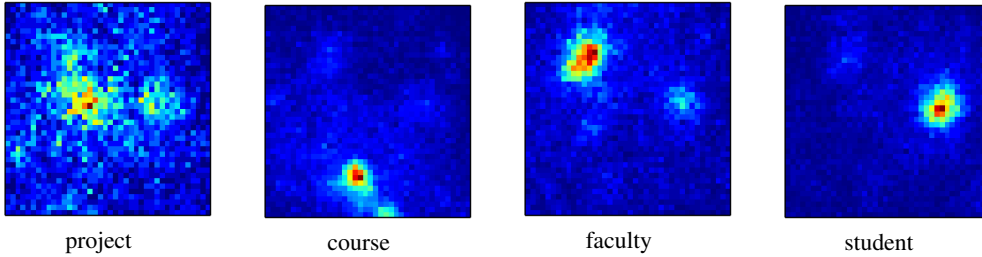

| project | course | faculty | student |

Figure 5: Distributions of latent vectors for words appearing in documents of each class on WebKB.

vectors, the latent SMM achieved high classification performance. On the other hand, MedLDA and SVD+SMM often could not display their own abilities when the latent dimensionality was low. One of the reasons why the latent SMM with a very low latent dimensionality $q$ achieves a good performance is that it can use $q|d_i|$ parameters to classify the $i$th document, while MedLDA uses only $q$ parameters. Since the latent word representation used in SVD+SMM is not optimized for the given classification problem, it does not contain useful features for classification, especially when the latent dimensionality is low.

**Accuracy over word occurrence threshold.** In the above experiments, we omit words whose occurrence accounts for less than 1% of the training document. By reducing the threshold, low frequency words become included in the training documents. This might be a difficult situation for the latent SMM and SVD+SMM because they cannot observe enough training data to estimate their own latent word vectors. On the other hand, it would be an advantageous situation for SVMs using BoW features because they can use low frequency words that are useful for classification to compute their kernel values. Figure 3 shows the classification accuracy on WebKB when varying the word occurrence threshold within $\{0.4, 0.6, 0.8, 1.0\}$. The performance of the latent SMM did not change when the thresholds were varied, and was better than the other methods in spite of the difficult situation.

**Parameter sensitivity.** Figure 4 shows how the performance of the latent SMM changes against $\ell_2$ regularizer parameter $\rho$ and $C$ on a Reuters-21578 dataset with 1,000 training samples. Here the latent dimensionality of the latent SMM was fixed at $q = 2$ to eliminate the effect of $q$. The performance is insensitive to $\rho$ except when $C$ is too small. Moreover, we can see that the performance is improved by increasing the $C$ value. In general, the performance of SVM-based methods is very sensitive to $C$ and kernel parameters [20]. Since kernel parameters $\theta$ in the latent SMM are estimated along with latent vectors $\mathbf{X}$, the latent SMM can avoid the problem of sensitivity for the kernel parameters. In addition, Figure 2 has shown that the latent SMM is robust over the latent dimensionality. Thus, the latent SMM can achieve high classification accuracy by focusing only on tuning the best $C$, and experimentally the best $C$ exhibits a large value, e.g., $C \geq 2^5$.

**Visualization of classes.** In the above experiments, we have shown that the latent SMM can achieve high classification accuracy with low-dimensional latent vectors. By using two- or three-dimensional latent vectors in the latent SMM, and visualizing them, we can understand the relationships between classes. Figure 5 shows the distributions of latent vectors for words appearing

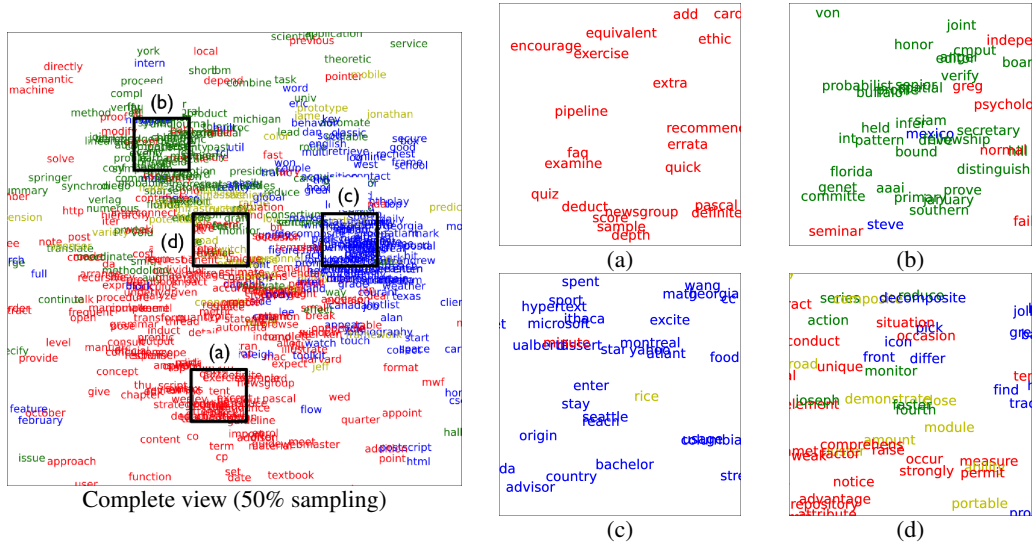

Complete view (50% sampling)

Figure 6: Visualization of latent vectors for words on WebKB. The font color of each word indicates the class in which the word occurs most frequently, and 'project', 'course', 'student' and 'faculty' classes correspond to yellow, red, blue and green fonts, respectively.

in documents of each class. Each class has its own characteristic distribution that is different from those of other classes. This result shows that the latent SMM can extract the difference between the distributions of the classes. For example, the distribution of 'course' is separated from those of the other classes, which indicates that documents categorized in 'course' share few words with documents categorized in other classes. On the other hand, the latent words used in the 'project' class are widely distributed, and its distribution overlaps those of the 'faculty' and 'student' classes. This would be because faculty and students work jointly on projects, and words in both 'faculty' and 'student' appear simultaneously in 'project' documents.

**Visualization of words.** In addition to the visualization of classes, the latent SMM can visualize words using two- or three-dimensional latent vectors. Unlike unsupervised visualization methods for documents, e.g., [21], the latent SMM can gather characteristic words of each class in a region. Figure 6 shows the visualization result of words on the WebKB dataset. Here we used the same learning result as that used in Figure 5. As shown in the complete view, we can see that highly-frequent words in each class tend to gather in a different region. On the right side of this figure, four regions from the complete view are displayed in closeup. Figures (a), (b) and (c) include words indicating 'course', 'faculty' and 'student' classes, respectively. For example, figure (a) includes 'exercise', 'examine' and 'quiz' which indicate examinations in lectures. Figure (d) includes words of various classes, although the 'project' class dominates the region as shown in Figure 5. This means that words appearing in the 'project' class are related to the other classes or are general words, e.g., 'occur' and 'differ'.

## 6 Conclusion

We have proposed a latent support measure machine (latent SMM), which is a kernel-based discriminative learning method effective for sets of features such as bag-of-words (BoW). The latent SMM represents each word as a latent vector, and each document to be classified as a distribution of the latent vectors for words appearing in the document. Then the latent SMM finds a separating hyperplane that maximizes the margins between distributions of different classes while estimating latent vectors for words to improve the classification performance. The experimental results can be summarized as follows: First, the latent SMM has achieved state-of-the-art classification accuracy for BoW data. Second, we have shown experimentally that the performance of the latent SMM is robust as regards its own hyper-parameters. Third, since the latent SMM can represent each word as a two- or three- dimensional latent vector, we have shown that the latent SMMs are useful for understanding the relationships between classes and between words by visualizing the latent vectors.

**Acknowledgment.** This work was supported by JSPS Grant-in-Aid for JSPS Fellows (259867).

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
