[Reviews · NeurIPS 2014]

Submitted by Assigned_Reviewer_38

The authors consider the problem of text classification using methods
beyond standard bag-of-word approaches. The basic technique is to
use an embedding of a document that incorporates latent
information and produces a sum of kernels representation. The latent
representation (hopefully) captures more semantically relevant
information. Buidling on top of the representation for documents,
classification is performed with a support measure machine. The
combination of the representation and the SMM is applied to standard
corpora and compared to other relevant approaches. Experiments show
the method needs less training data, produces good results with low
dimensional latent representations, and is interpretable
qualitatively.

Overall, this paper is a nice approach to document classification
based upon recent latent mappings for words in documents. The
experiments show the potential of the approach. The paper is clear
and the method is straightforward once the authors motivate and
describe the approach. The experimental section is especially
complete with a set of reasonable experiments and qualitative
interpretations in Figure 6.

A few minor suggestions might help the paper. Can the authors relate
their approach to standard terminology. For instance, is the
embedding in (2) equivalent to a non-parameteric distribution
estimator? Also, the kernel in (3) looks like an L^1 kernel.
Summary: The authors present methods for mapping document to latent
representations which are then used in a support measure machine.
The approach is convincing and has good potential for interesting
follow on research.

Submitted by Assigned_Reviewer_42

This paper proposes a novel method for performing classification (or
other similar tasks) on bag-of-words data. The main idea is to learn
a classifier and feature representation simultaneously. The
classifier is a support measure machine (SMM). The feature
representation is a latent vector for each word in the vocabulary.
The method thus resembles approaches like one proposed by Muandet and
Fukumizu in NIPS 2012, which uses an unsupervised embedding procedure
together with SMMs, except that in the current work both tasks are
formulated as a single optimization.

The manuscript is very clear. The results are strong. The core idea
is interesting though not particularlyexciting. Instead, this seems
like a well motivated but fairly straightforward extension of existing
methods.

In the empirical results, it would be nice to increase the
hyperparameter space. The results shown in Figure 4 suggest that the
parameter \rho should be varied more substantially, and that larger
values of C should be included.

The experiment concerning accuracy over word occurrence threshold is
not informative. Essentially, the results don't change for any
method. A wider range of thresholds should be used, or this
experiment should be replaced with one that is more informative.

The use of English is imperfect in many places. Careful editing is
required. E.g., in the abstract "we shows that" should be "we show
that." Some phrasing is also awkward, e.g., "closely located each
other in the latent space."

Summary: This manuscript presents a compelling core idea -- to formulate the
learning of latent embeddings for bag-of-words model jointly with the
learning of a discriminative classifier -- coupled with well executed
experiments and reasonably strong results.

Submitted by Assigned_Reviewer_44

Latent Support Measure Machines for Bag-of-Words Data Classification

This paper presents a kernel-based latent support measure machine for bag-of-word data to jointly learn word embedding and a document classifier. It assumes each word can be represented as a latent vector and each document is represented as a distribution of the latent vectors associated with the words in the document. Then latent support measure machine learns a model to classify documents based on the latent distribution. The proposed method is a nice extension of support measure machines. Experiments show that the proposed method outperforms other methods using word embedding or word cluster. The paper is nice written and easy to follow.

Comments:
- My main concern is the scalability of the proposed method. All the experiments are conducted using less than 1,000 examples. However, it is easy to encounter a bag-of-word data in the scale of hundred thousand in text classification problems. It is unclear if the proposed method can scale up to deal with such datasets.
- The paper demonstrates promising results using a small dataset. However, it is hard to judge if Latent SMM can improve the accuracy of text classification when the training samples are sufficient. For example, new20 achieves 0.82 accuracy using SVM with linear kernel when training on the whole dataset (http://web.ist.utl.pt/acardoso/datasets/). The accuracy of the best method trained on 1,000 samples reported in Figure 1 (c) is around 0.6. It would be more promising to report the results on the whole dataset as well.
- I wonder if the proposed method can be extended for other applications.
- In figures 4, it seems that the method performs better when using a large C to fit training data. Is there an explanation for that?

Minor comments/typos:
- Lines 361, 364: should be Figure 3, Figure 4, respectively.

I like the experimental analysis about the word occurrence threshold (line 356~363). However, it is hard to see the performance difference with different thresholds in Figure 3.
Summary: This paper is well motivated and well written and provides a nice experimental analysis. However, it is not clear if the proposed method can scale up and can perform well on large datasets.
Author Feedback
Author rebuttal: We would like to thank the reviewers for their feedback and insightful comments, which we shall address below.

--
[Assigned_Reviewer_38]

> is the embedding in (2) equivalent to a non-parameteric distribution estimator?

The kernel embedding in Eq. (2) is not equivalent to a non-parametric distribution estimator using kernel density estimation (KDE), although these formulations are similar.
The kernel embedding represents the moment information (e.g., mean, covariance and higher-order moments) of a distribution, rather than its density.

--
[Assigned_Reviewer_42]

> In the empirical results, it would be nice to increase the hyperparameter space. ...

As you suggested, further experiments with increasing the hyperparameter space make the property of the proposed method more clear.
In the final version of the paper, we will add the results of the experiments.

> The experiment concerning accuracy over word occurrence threshold is not informative. ...

When word occurrence threshold is low, low frequency words become included in the training documents. Therefore, overfitting might occur for latent word vectors of low frequency words. However, Figure 3 shows that the performance of the proposed method does not change when varying the word occurrence threshold.

> The use of English is imperfect in many places. ...

In the final version, we will improve the quality of the paper by asking for proofreading.

--
[Assigned_Reviewer_44]

> My main concern is the scalability of the proposed method. ...

The most computationally expensive part in learning is the estimation of latent vectors for words. For the computation of this part, we can employ stochastic gradient decent, which can be calculated with O(W^2) of time complexity for each word vector, where W is the average number of words in a document.

> The paper demonstrates promising results using a small dataset. ...

A strong point of the proposed method is its high classification performance even with small training data.
Figure 1 shows the accuracy of the proposed method improves with increasing the number of training documents, and for each of data set sizes the proposed method achieved high accuracy compared with other methods.

> I wonder if the proposed method can be extended for other applications.

The proposed method can be applied to various tasks, such as novelty detection, structure prediction and learning to rank, as described in lines 83-86

> In figures 4, it seems that the method performs better when using a large C to fit training data. Is there an explanation for that?

Since a large C leads to a hard-margin classifier, the proposed method learns latent word vectors so as to classify documents under the hard-margin principle.
A reason why the proposed method can avoid over-fitting even with hard-margin would be because it can measure kernels between documents robustly by representing each document as a distribution of word vectors.

> Minor comments/typos: Lines 361, 364: should be Figure 3, Figure 4, respectively.

We will fix those in the final version of the paper.

> I like the experimental analysis about the word occurrence threshold (line 356‾363). However, it is hard to see the performance difference with different thresholds in Figure 3.

The performance does not change when we change the word occurrence threshold. This result indicates that the proposed method does not overfit even when the vocaburary size is large and low frequency words are included in the training data.